

# Effect of arbuscular mycorrhizal symbiosis on growth and biochemical characteristics of Chinese fir (*Cunninghamia lanceolata*) seedlings under low phosphorus environment

Yunlong Tian[1,2], Jingjing Xu[1,2], Linxin Li[1,2], Taimoor Hassan Farooq[3], Xiangqing Ma[1,2] and Pengfei Wu[1,2]

[1] College of Forestry, Fujian Agriculture and Forestry University, Fuzhou, Fujian, China
[2] Chinese Fir Engineering Technology Research Center of the State Forestry and Grassland Administration, Fuzhou, Fujian, China
[3] Bangor College, Central South University of Forestry and Technology, Changsha, Hunan, China

Corresponding author
Pengfei Wu, fjwupengfei@126.com

## ABSTRACT

**Background**. The continuous establishment of Chinese fir (*Cunninghamia lanceolata*) plantations across multiple generations has led to the limited impact of soil phosphorus (P) on tree growth. This challenge poses a significant obstacle in maintaining the sustainable management of Chinese fir.

**Methods**. To investigate the effects of Arbuscular mycorrhizal fungi (AMF) on the growth and physiological characteristics of Chinese fir under different P supply treatments. We conducted an indoor pot simulation experiment in the greenhouse of the Forestry College of Fujian Agriculture and Forestry University with one-and-half-year-old seedlings of Chinese fir from March 2019 to June 2019, with the two P level treatment groups included a normal P supply treatment ($1.0$ mmol $L^{-1}$ $KH_2PO_4$, P1) and a no P supply treatment ($0$ mmol $L^{-1}$ $KH_2PO_4$, P0). P0 and P1 were inoculated with *Funneliformis mosseae* (*F.m*) or *Rhizophagus intraradices* (*R.i*) or not inoculated with AMF treatment. The AMF colonization rate in the root system, seedling height (SH), root collar diameter (RCD) growth, chlorophyll (Chl) photosynthetic characteristics, enzyme activities, and endogenous hormone contents of Chinese fir were estimated.

**Results**. The results showed that the colonization rate of *F.m* in the roots of Chinese fir seedlings was the highest at P0, up to 85.14%, which was 1.66 times that of P1. Under P0 and P1 treatment, root inoculation with either *F.m* or *R.i* promoted SH growth, the SH of *R.i* treatment was 1.38 times and 1.05 times that of *F.m* treatment, respectively. In the P1 treatment, root inoculation with either *F.m* or *R.i* inhibited RCD growth. *R.i* inhibited RCD growth more aggressively than *F.m*. In the P0 treatment, root inoculation with *F.m* and *R.i* reduced the inhibitory effect of phosphorus deficiency on RCD. At this time, there was no significant difference in RCD between *F.m*, *R.i* and *CK* treatments ($p < 0.05$). AMF inoculation increased *Fm*, *Fv*, *Fv/Fm*, and *Fv/Fo* during the chlorophyll fluorescence response in the tested Chinese fir seedlings. Under the two phosphorus supply levels, the trend of *Fv* and *Fm* of Chinese fir seedlings in different treatment groups was *F.m* > *R.i* > CK. Under P0 treatment, The values of *Fv* were 235.86, 221.86 and 147.71, respectively. The values of *Fm* were 287.57, 275.71 and 201.57, respectively.

It increased the antioxidant enzyme activity and reduced the leaf's malondialdehyde (MDA) content to a certain extent.

**Conclusion**. It is concluded that AMF can enhance the photosynthetic capacity of the host, regulate the distribution of endogenous hormones in plants, and promote plant growth by increasing the activity of antioxidant enzymes. When the P supply is insufficient, AMF is more helpful to plants, and *R.i* is more effective than *F.m* in alleviating P starvation stress in Chinese fir.

## INTRODUCTION

Arbuscular mycorrhizal fungi (AMF) can form mutually beneficial symbioses with most terrestrial plants and are important partners in the long-term natural evolution of plants. AMF colonizes the plant root system and transports nitrogen (N), phosphorous (P), and water through the extra-root mycelium, ensuring plants' normal growth under nutrient-deficient conditions (*Smith, Smith & Jakobsen, 2004*; *Jiang et al., 2021*). In exchange for acquiring resources, plants reciprocate by supplying carbon (C) to AMF, contributing approximately 4% to 25% of the total plant photosynthetic output (*Zhou et al., 2020*; *Kaur, Campbell & Suseela, 2022*). Improvements in plant P nutrient availability by AMF will inevitably directly impact plant leaf photosynthesis and metabolic responses (*Schweiger, Baier & Müller, 2014*; *Jajoo & Mathur, 2021*).

Studies have shown that AMF can regulate chloroplast enzyme activity, accelerate the synthesis of essential enzymes required for the chlorophyll peptide chain, promote chlorophyll synthesis, and increase the chlorophyll content of plants while decreasing the rate of chlorophyll decomposition to increase the intensity of photosynthesis and improve the efficiency of nutrient uptake by the root system (*Eulenstein et al., 2016*). In addition, AMF inoculation has a protective effect on beach plum (*Prunus maritima*) photosystem II (PSII), which can increase the efficiency of light energy conversion and improve the original response of photosynthesis under salt stress (*Zai et al., 2012*). Although improved P nutrition is undoubtedly one of the main effects of AMF on plants, only a few studies have shown whether changes in plant growth, metabolites, and photosynthesis are driven exclusively by this nutritional effect. It is possible that, in addition to its basal function of improving the supply of nutrients to the plant, AMF also responds to environmental stresses by modulating the secretion of secondary metabolites and endogenous hormones in the plant's root system (*Schweiger, Baier & Müller, 2014*).

P is essential for plant growth, is an integral part of plant nucleic acid structures and biofilms, is involved in the biosynthesis of a wide range of primary and secondary metabolites, and plays a vital role in cell division and tissue development (*Karandashov & Bucher, 2005*). The adaptive response of plants to low P stress not only forms a series of adaptive mechanisms at the morpho-anatomical level but also induces physiological and
biochemical changes in the plant in terms of increased root secretion, antioxidant enzyme activity, leaf photosynthetic pigments, and endogenous hormones, which give the plant an induced ability to adapt to low P to acquire P (*Ciereszko, Szczygła & Żebrowska, 2011*; *Lin et al., 2012*; *Zubek, Mielcarek & Turnau, 2012*; *Desai, Naik & Cumming, 2014*; *Liu et al., 2014*).

Antioxidant enzymes are important in plant defense against oxidative stress induced by various biotic and abiotic factors. Under low P stress, plants can inhibit the formation of MDA by altering the activities of antioxidant enzymes, such as superoxide dismutase (SOD), catalase (CAT), and peroxidase (POD), and reduce lipid membrane peroxidation, thereby mitigating the damage to the cell membrane system (*Hanin et al., 2016*). In addition, environmental stresses also inhibit plant photosynthesis to a certain extent by affecting plant physiology and metabolism, thus reducing photosynthetic C fixation and inhibiting plant growth (*Desai, Naik & Cumming, 2014*; *Chitarra et al., 2016*). As a class of organic substances produced by the plant's metabolism, phytohormones are essential in regulating plant growth, development, and differentiation, and they are important signaling substances for plant adaptation to adversity (*Moore, 1985*; *He et al., 2020*). Plants often adapt to environmental stress under stress conditions through endogenous hormone responses in the form of modulation of plant growth rhythms, root growth, and increased protective enzyme activities (*Llanes et al., 2016*; *Etesami, Jeong & Glick, 2021*).

Chinese fir (*Cunninghamia lanceolata*) has been widely planted in tropical and subtropical mountainous regions because of its excellent growth characteristics and industrial and commercial uses, and it is a crucial afforestation tree species commonly used in China (*Zhao et al., 2009*). In subtropical and tropical regions, most of the P required for plant growth comes from two processes: P reabsorption before leaf abscission and P mineralization by microorganisms. Especially for evergreen coniferous forests, which have poor nutrient regression due to the species' biological characteristics, the limiting effect of P on forest growth and development may be more severe for coniferous forest species such as Chinese fir (*Fang et al., 2017*; *Pan et al., 2023*). During long-term natural selection, Chinese fir has evolved a variety of morphological and physiological adaptive mechanisms to cope with environmental P stress, either through morphological changes in the root system to enhance P foraging or through secretion of root secondary metabolites and cortical solubilization to improve its P utilization efficiency; however, such adaptive mechanisms are generally accomplished by increasing the nutrient-biased investment of the plant in its specific function, which may affect the maintenance and enhancement of the sustainable productivity of the forest (*Wu et al., 2018*; *Zou et al., 2018*).

Recent studies have shown that Chinese fir can achieve a better symbiotic relationship with AMF through its roots. This interaction aids in enhancing the tree's ability to withstand nutrient stress, thereby contributing to the sustainable management of both economic and ecological benefits within Chinese fir plantations (*Li et al., 2019*). In addition, some studies have shown that different species of AMFs have different functions in assisting plants and that AMFs illustrate different interactions in host habitats with varying concentrations of nutrients (*Liu et al., 2014*; *Xie et al., 2014*; *Qin et al., 2017*; *Shi, Wang & Wang, 2023*). Then, how does the growth stress response process of Chinese fir behave after AMF forms

a symbiotic relationship with Chinese fir? Are the results of the growth effects of different species of AMF on Chinese fir also related to environmental nutrient concentrations? There are fewer reports of related studies in this area.

We hypothesized that (1) AMF inoculation of Chinese fir roots is postulated to enhance Chinese fir growth. The symbiotic association between AMF and Chinese fir roots is believed to enhance nutrient absorption capacity and stress resistance, thereby fostering growth. Specifically, the promoting effect of AMF on Chinese fir growth may be heightened in P-deficient conditions, which often limit plant growth. The mycorrhizal formation between AMF and Chinese fir roots is expected to expand the root absorption area, aiding in more effective utilization of limited P resources. (2) The promoting effect of various AMF species on Chinese fir growth could be correlated with P concentration. Different AMF types could exert distinct influences on plant growth. Certain AMF species may play a more significant role in promoting Chinese fir growth in soil conditions with ample P, as the tree can readily access sufficient P. Conversely, other AMF species might exhibit stronger growth-promoting effects in soil with lower phosphorus concentrations, given their superior phosphorus acquisition and transport capabilities. (3) AMF is presumed to impact the endogenous hormone levels of Chinese fir roots and aboveground plants, thereby regulating overall plant growth and development. AMF is expected to stimulate the production of indole-3-acetic acid (IAA), a crucial plant hormone that promotes root growth and differentiation. Additionally, in P-deficient conditions, AMF is anticipated to enhance the activity of various enzymes in Chinese fir, augmenting antioxidant capacity and stress resistance. (4) AMF is postulated to enhance the fluorescence characteristics of Chinese fir chloroplasts. In P-deficient conditions, AMF is believed to increase chlorophyll fluorescence reactions in Chinese fir chloroplasts, improving photosynthetic efficiency and ultimately promoting Chinese fir's overall growth and development.

## MATERIAL AND METHODS

### Plant materials and growth conditions

Eighteen-month-old Chinese fir seedlings of the same clone (No.41) were chosen as test materials with a mean seedling height of $19.1 \pm 0.5$ cm and a root collar diameter (RCD) of $3.03 \pm 0.08$ mm, which were robust and free of pests and diseases. The Chinese Fir Engineering Technology Research Center of the State Forestry and Grassland Administration cultivated these seedlings. The chosen seedlings exhibit high nutritional requirements, strong adaptability, and moderate growing periods to be suitable for reflecting the actual situation and drawing accurate and reliable research conclusions (*Tian et al., 2023*). We conducted experimental research from March 2019 to June 2019, all the seedlings were planted in polyethylene pots in a glasshouse at the Forestry College, Fujian Agriculture and Forestry University. The growing medium in each pot was filled with 5.0 kg of the mixed substrate with cultivation substrate and fungal soil, following the volume ratio of 6:0.6. The cultivation substrate was a mixture of river sand and perlite (3:1). Before utilization, the high-pressure steam method was used to sterilize the cultivation substrate for a duration 30 min (121 °C, 0.1~0.2 MPa).

The fungal soil containing the corresponding matrix, AMF spores, and extraradical hyphae was provided by the Institute of Plant Nutrition and Resources, Beijing Academy of Agricultural and Forestry Sciences. Two kinds of AMF, which are often used to inoculate plant roots and are more easily symbiotic with plants, were selected. *Funneliformis mosseae* (*F.m*) and *Rhizophagus intraradices* (*R.i*) were added into the pot as the experimental material, with no AMF supplied as the control treatment (CK) (*Schüßler & Walker, 2010*; *Walker et al., 2021*).

The pH values of the mixed substrate made from *F.m*, *R.i* and CK were $6.33 \pm 0.12$, $6.52 \pm 0.15$ and $6.59 \pm 0.07$, respectively, and the available P concentration was $0.21 \pm 0.03$ mg kg$^{-1}$, $0.32 \pm 0.05$ mg kg$^{-1}$ and $0.29 \pm 0.02$ mg kg$^{-1}$. The growing condition in the greenhouse was 18–28 °C; the average photoperiod was 10 h day$^{-1}$, and relative humidity >80% during the experiment.

## Experimental methods

According to the previous test methods, the experimental instruments and Chinese fir seedlings were disinfected and sterilized to reduce exogenous AMF infestation (*Tian et al., 2023*). Two P supply level treatment groups were set up, each with three AMF inoculation treatments of *F.m* or *R.i* and no AMF, respectively. Regarding our previous research methods, the two P levels treatment groups included a normal P supply treatment (1.0 mmol L$^{-1}$KH$_2$PO$_4$, P1) and a no P supply treatment (0 mmol L$^{-1}$KH$_2$PO$_4$, P0). Completely randomized design (CRD) with seven replicates for each treatment was used; there were 42 pots in total.

Each pot was supplied with a quarter of the modified Hoagland nutrient solution formula to satisfy the tested seedling requirements for other nutrients—each time with 60 ml every 7 days. And 200 ml of pure water was poured every 5 days in the afternoon. The detailed implementation method refers to our previous research (*Tian et al., 2023*). The tested seedlings were harvested after 90 days.

## Harvest and data collection

Before harvesting, the origin fluorescence (*Fo*), maximum fluorescence (*Fm*) and maximum photochemical efficiency of PS II (*Fv/Fm*) of the photochemical reaction were determined by using a chlorophyll (chl) fluorometer (OS-30P, Li-Cor) to select mature leaves of the participant seedlings on a sunny day from 09:00 to 14:00 and calculate the photochemical reaction variable fluorescence (*Fv*) and potential photochemical activity of PS II (*Fv/Fo*) were also calculated. Chinese fir seedling height and root collar diameter were measured before and after the experimental treatments using a steel ruler (accuracy of 0.1 cm) and vernier calipers (accuracy of 0.01 mm), respectively. The growth of Chinese fir seedling height and root collar diameter were calculated after comparing the measured values of seedling height and root collar diameter before and after the experimental treatments.

The root staining procedure was modified from the Trypan Blue method (*Phillips & Hayman, 1970*). The decolorized root segments were observed using a light microscope. Whenever mycorrhizal structures such as hyphae, vesicles, and arbuscule appeared in the root segments, the roots were considered colonized by AMF. The seedlings of Chinese fir

not inoculated with AMF were examined under a microscope, and no mycelia were found in the observed root segments. The AMF colonization rate (F/%) equals the number of root segments infested with mycorrhizal as a percentage of all observed root segments (*Xie et al., 2014*).

At the end of the experiment, when the seedlings were harvested, 0.20 g of leaves and newborn roots were weighed, washed, and dried with a cotton cloth, then cut into five mL centrifuge tubes, sealed and frozen in liquid N, then crushed into a homogenate with a tissue grinder, and then frozen at 10,000 r min$^{-1}$ at 4 °C with four mL of Phosphate buffer solution (PBS) (0.05 mol L$^{-1}$, pH = 7.8) added. After centrifugation for 20 min at 4 °C for 10,000 rpm, the supernatant was the enzyme solution to be measured. The superoxide dismutase (SOD) activity of Chinese fir leaves and roots was determined by the photochemical reduction method of nitrogen blue tetrazolium, the catalase (CAT) activity was determined by ultraviolet absorption method, and the malondialdehyde (MDA) content was determined by thiobarbituric acid method (*Li, 2000*). Using Enzyme-Linked Immunosorbent Assay (ELISA) (*Sakamoto et al., 2018*) to determine the content of endogenous hormones growth hormone (IAA), abscisic acid (ABA), and zeatin riboside (ZR) in Chinese fir leaves and root system, each hormone kit was provided by Shanghai Kexing Trading Co., Ltd. under the brand name of Fankew.

### Statistical analyses

Two-way ANOVA and single-factor analysis of variance (One-way ANOVA) were performed on the experimental data using SPSS 25.0 and Duncan's multiple comparison method ($p = 0.05$). All data were expressed as mean ± standard error (SE) and Pearson correlation analysis, and correlation charts were drawn using Origin 2021.

## RESULTS

### Changes in seedling colonization rate of *F.m* in the roots

According to the effects of different phosphorus supply levels on the colonization rate of *F.m* in the roots of Chinese fir seedlings (Fig. 1), it can be seen that the colonization rate of *F.m* in the roots in P0 treatment was higher than that in P1 treatment ($p < 0.05$). Among them, at P0, the colonization rate of *F.m* in the roots of Chinese fir seedlings was the highest, up to 85.14%, which was 1.66 times that of P1. Our previous study found that the colonization rate of *R.i* in the roots in P0 treatment was higher than that in P1 treatment ($p < 0.05$). And the colonization rate of *R.i* in the roots of Chinese fir seedlings was the highest at P0, up to 69.81%, which was 1.47 times that of P1 (*Tian et al., 2023*).

### Changes in seedling height and root collar diameter

From the results of two-way ANOVA (Table 1), it can be seen that the two factors, P supply level, and AMF inoculation, had a significant interaction effect ($p < 0.01$) on the RCD of Chinese fir. In addition, the AMF inoculation treatment has a significant impact ($p < 0.01$) on Chinese fir SH and RCD. According to the effects of AMF inoculation on SH and RCD of Chinese fir under different P levels (Fig. 2), under P1, the trend of SH size of Chinese fir seedlings in different treatment groups was *R.i* > *F.m* > CK, which were 19.84, 18.93

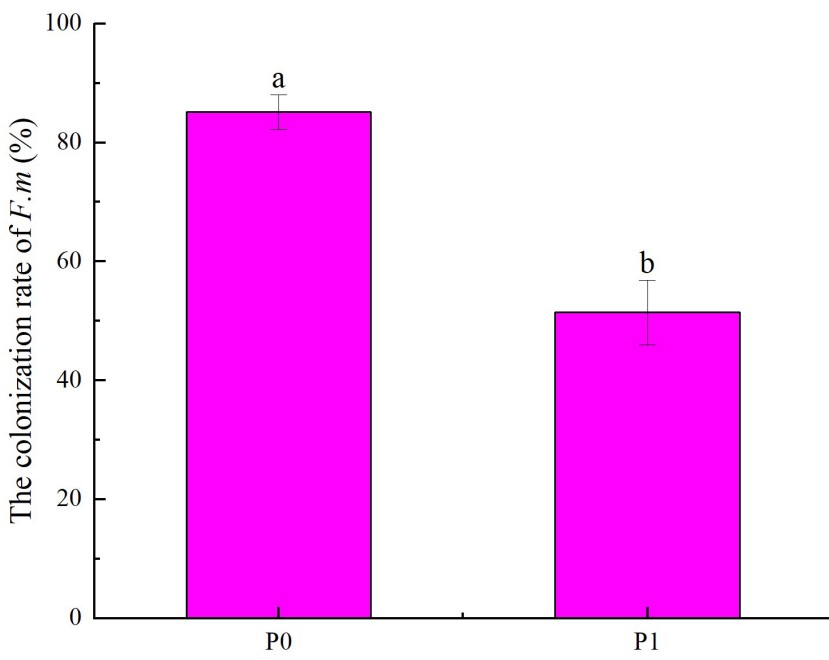

**Figure 1** **Effects of different phosphorus supply levels on the colonization rate of *F.m* in the roots of Chinese fir seedlings.** In the figure, P0 represents no phosphorus treatment and P1 represents normal phosphorus treatment. *F.m* represents treatment with *Funneliformis mosseae* inoculation. Different lower-case letters represent significant differences between the two treatments ($p < 0.05$).

and 16.66, respectively. The trend of RCD of Chinese fir seedlings in different treatment groups was CK > *F.m* > *R.i*, 3.26, 1.64 and 1.51, respectively. Chinese fir roots inoculated with *F.m* or *R.i* promoted SH, but the RCD was significantly lower than that of the CK treatment ($p < 0.05$); in the P0 treatment, the trend of SH size of Chinese fir seedlings in different treatment groups was *R.i* > *F.m* > CK, which were 25.77, 18.73 and 16.81, respectively. The trend of RCD of Chinese fir seedlings in different treatment groups was CK > *R.i* > *F.m*, 2.033, 2.029, and 1.989, respectively. Root inoculation with *F.m* and *R.i* promoted the SH of Chinese fir and reduced the inhibitory effect of phosphorus deficiency on RCD. Among them, the inoculation of *R.i* significantly promoted the SH of Chinese fir ($p < 0.05$).

## Changes in chlorophyll photochemical properties in seedling leaves

Two factors of P supply level and AMF inoculation showed significant interaction ($p < 0.01$) on maximum photochemical efficiency of PSII ($Fv/Fm$) and potential photochemical activity of PSII ($Fv/Fo$) in Chinese fir leaves (Table 2). In addition, the treatment of AMF inoculation had significant effects ($p < 0.01$) on variable fluorescence ($Fv$), $Fv/Fm$, $Fv/Fo$, and Chl a/b of Chinese fir leaves. P supply level significantly affects ($p < 0.01$) on Chl a/b.

Based on the effects of AMF inoculation on Chl photochemical characteristics of Chinese fir leaves under different P levels (Fig. 3), maximum fluorescence ($Fm$), $Fv$, $Fv/Fm$, and $Fv/Fo$ of Chinese fir were higher than those of the CK treatment at different P supply treatments after root inoculation with *F.m* or *R.i*. Under the two phosphorus supply levels,

**Table 1 Two-way ANOVA of the effects of phosphorus supply level and AMF inoculation on the growth of SH and RCD of Chinese fir.**

| Index | Factor | SS | df | MS | F | p |
|---|---|---|---|---|---|---|
| SH | P supply level (a) | 40.415 | 1 | 40.415 | 2.500 | 0.123 |
| RCD | | 0.146 | 1 | 0.146 | 0.306 | 0.584 |
| SH | AMF inoculation (b) | 266.333 | 2 | 133.166 | 8.237** | 0.001 |
| RCD | | 6.819 | 2 | 3.410 | 7.122** | 0.002 |
| SH | a × b | 82.829 | 2 | 41.415 | 2.562 | 0.091 |
| RCD | | 6.473 | 2 | 3.237 | 6.761** | 0.003 |
| SH | Error | 582.026 | 36 | 16.167 | | |
| RCD | | 17.234 | 36 | 0.479 | | |
| SH | Total | 16871.980 | 42 | | | |
| RCD | | 211.633 | 42 | | | |

Notes.

SH represents seedling height increment and RCD represents root collar diameter increment.

Asterisks (* and **) represent the influence of factors on the indicators reached a significant difference level at $p < 0.05$ and $p < 0.01$, respectively.

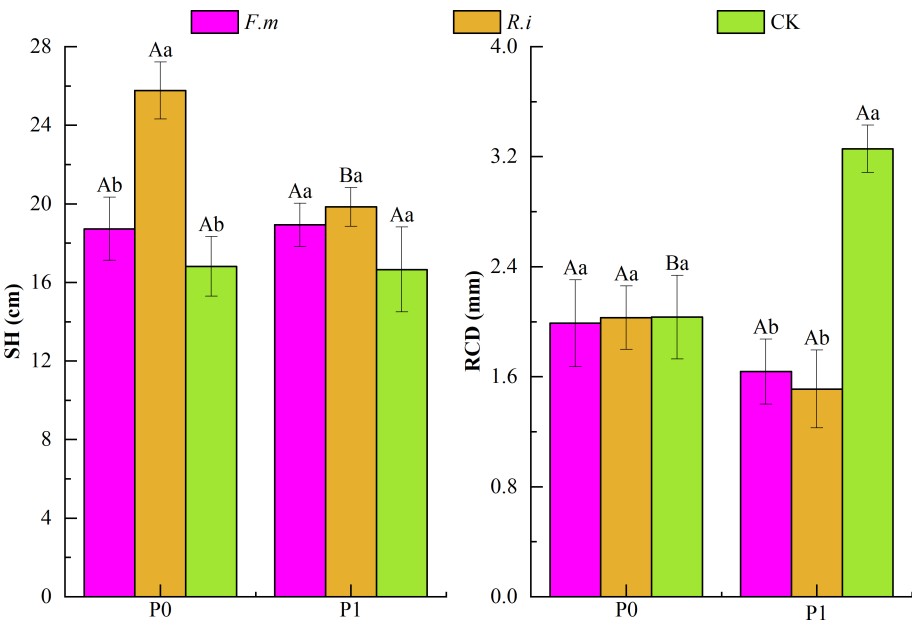

**Figure 2 Effects of Arbuscular mycorrhizal fungi (AMF) inoculation on seedling height (SH) and root collar diameter (RCD) growth of Chinese fir under different phosphorus levels.** In the figure, P0 represents no phosphorus treatment, P1 represents normal phosphorus treatment; *F.m* represents treatment with *Funneliformis mosseae* inoculation, *R.i* represents treatment with *Rhizophagus intraradices* inoculation, and CK represents the no inoculation treatment. SH represents seedling height increment and RCD represents root collar diameter increment. Different capital letters represents a significant differences between different phosphorus treatments under same AMF inoculation treatment. Whereas different lowercase letters represent significant differences between different AMF inoculation treatments when the phosphorus supply treatment is the same ($p < 0.05$).

the trend of *Fv* and *Fm* of Chinese fir seedlings in different treatment groups was *F.m* > *R.i* > CK. Under P1 treatment, the values of *Fv* were 194.29, 181.43, and 161.14, respectively. The values of *Fm* were 240.14, 235.29, and 213.14, respectively. However, at P0 treatment,

**Table 2  Two-way ANOVA of the effects of phosphorus supply level and arbuscular mycorrhizal fungi (AMF) inoculation on chlorophyll photochemical characteristics of Chinese fir leaves.**

| Index | Factor | SS | df | MS | F | p |
|---|---|---|---|---|---|---|
| Fo | | 69.429 | 1 | 69.429 | 0.580 | 0.451 |
| Fm | | 6789.429 | 1 | 6789.429 | 2.667 | 0.111 |
| Fv | P supply level (a) | 5485.714 | 1 | 5485.714 | 3.334 | 0.076 |
| Fv/Fm | | 0.000 | 1 | 0.000 | 0.750 | 0.392 |
| Fv/Fo | | 0.400 | 1 | 0.400 | 2.505 | 0.122 |
| Chl a/b | | 0.322 | 1 | 0.322 | 8.631[**] | 0.006 |
| Fo | | 204.143 | 2 | 102.071 | 0.852 | 0.435 |
| Fm | | 26039.190 | 2 | 13019.595 | 5.114[*] | 0.011 |
| Fv | AMF inoculation (b) | 28406.333 | 2 | 14203.167 | 8.633[**] | 0.001 |
| Fv/Fm | | 0.041 | 2 | 0.020 | 44.641[**] | 0.000 |
| Fv/Fo | | 16.286 | 2 | 8.143 | 50.959[**] | 0.000 |
| Chl a/b | | 0.476 | 2 | 0.238 | 6.372[**] | 0.004 |
| Fo | | 62.714 | 2 | 31.357 | 0.262 | 0.771 |
| Fm | | 7273.000 | 2 | 3636.500 | 1.428 | 0.253 |
| Fv | a × b | 6914.714 | 2 | 3457.357 | 2.101 | 0.137 |
| Fv/Fm | | 0.008 | 2 | 0.004 | 8.203[**] | 0.001 |
| Fv/Fo | | 2.573 | 2 | 1.287 | 8.051[**] | 0.001 |
| Chl a/b | | 0.165 | 2 | 0.083 | 2.210 | 0.124 |
| Fo | | 4312.857 | 36 | 119.802 | | |
| Fm | | 91650.000 | 36 | 2545.833 | | |
| Fv | Error | 59229.143 | 36 | 1645.254 | | |
| Fv/Fm | | 0.016 | 36 | 0.000 | | |
| Fv/Fo | | 5.753 | 36 | 0.160 | | |
| Chl a/b | | 1.345 | 36 | 0.037 | | |
| Fo | | 117594.000 | 42 | | | |
| Fm | | 2596282.000 | 42 | | | |
| Fv | Total | 1622322.000 | 42 | | | |
| Fv/Fm | | 25.712 | 42 | | | |
| Fv/Fo | | 602.808 | 42 | | | |
| Chl a/b | | 196.970 | 42 | | | |

**Notes.**

*Fo* represents origin fluorescence, *Fv* represents variable fluorescence, *Fm* represents maximum fluorescence, *Fv/Fm* represents maximal photochemical efficiency of PSII, *Fv/Fo* represents the potential photochemical activity of PSII, and Chl a/b represents the ratio of chlorophyll-a content to chlorophyll b content.

\* and \*\* represent the influence of factors on the indicators reached a significant difference at $p < 0.05$ and $p < 0.01$, respectively.

the values of *Fv* were 235.86, 221.86 and 147.71, respectively. The values of *Fm* were 287.57, 275.71 and 201.57, respectively.

In particular, leaf *Fv/Fm* and *Fv/Fo* were significantly higher ($p < 0.05$) than the CK treatments after root inoculation with *F.m* or *R.i* in the P0 treatments in Chinese fir. *Fv* and *Fm* were also significantly higher ($p < 0.05$) than the CK treatments after root inoculation with *F.m* or *R.i*, respectively. However, the origin fluorescence (*Fo*) of Chinese fir was not significantly different ($p > 0.05$) from the CK after root inoculation with *F.m* or *R.i* at different P supply treatments. After root inoculation with *R.i* at P0 level, the fluorescence

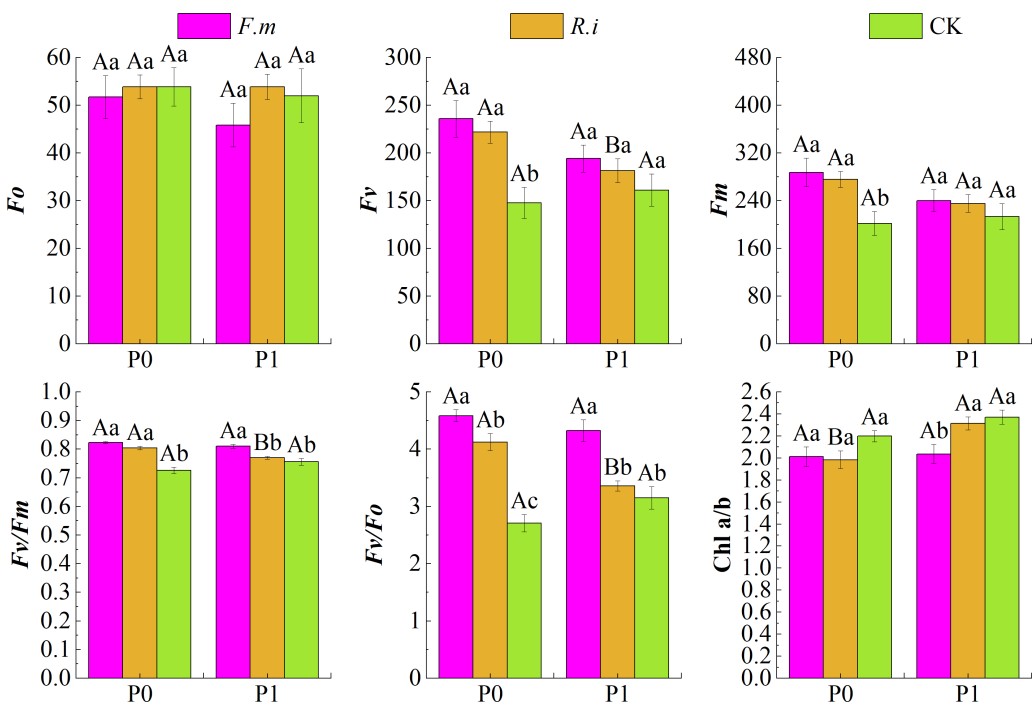

**Figure 3** **Effects of AMF inoculation on chlorophyll (Chl) photochemical characteristics of Chinese fir leaves under different phosphorus levels.** In the figure, P0 represents no phosphorus treatment, P1 represents normal phosphorus treatment; *F.m* represents treatment with *Funneliformis mosseae* inoculation, *R.i* represents treatment with *Rhizophagus intraradices* inoculation, and CK represents the no inoculation treatment. *Fo* represents origin fluorescence, *Fv* represents variable fluorescence, *Fm* represents maximum fluorescence, *Fv/Fm* represents maximal photochemical efficiency of PSII, *Fv/Fo* represents the potential photochemical activity of PSII and Chl a/b represents the ratio of chlorophyll content to chlorophyll b content. Different capital letters represents a significant differences between different phosphorus treatments under same AMF inoculation treatment. Whereas different lowercase letters represent significant differences between different AMF inoculation treatments when the phosphorus supply treatment is the same ($p < 0.05$).

parameters *Fv*, *Fv/Fm* and *Fv/Fo* of Chinese fir were significantly improved compared with those at P1 level. In addition, Chinese fir had lower leaf Chl a/b after root inoculation with *F.m* or *R.i* at different P supply treatments than the CK treatment. Under P1 treatment, the trend of Chl a/b in different treatment groups was CK > *R.i* > *F.m*, and the values of Chl a/b were 2.37, 2.31, and 2.04, respectively. At P0 treatment, the trend of Chl a/b in different treatment groups was CK > *F.m* > *R.i*, and the values of Chl a/b were 2.20, 2.01, and 1.98, respectively.

## Changes in antioxidant enzyme activities in seedling leaves and roots

P supply level, AMF inoculation, and their interaction showed a significant effect on ($p < 0.01$) for CAT and SOD activity in Chinese fir roots and leaves (Table 3). Data in Fig. 4 shows that Chinese fir leaves had lower MDA content than the CK after root inoculation with *F.m* or *R.i* in the P0 treatments, but root inoculation with *F.m* or *R.i* did not have a significant effect on the changes in root MDA content ($p > 0.05$). In contrast, Chinese fir

**Table 3  Two-way ANOVA of the effects of phosphorus supply level and arbuscular mycorrhizal fungi (AMF) inoculation on the enzyme activities in leaves and roots of Chinese fir.**

| Index | Factor | SS | df | MS | F | p |
|---|---|---|---|---|---|---|
| L_MDA | | 0.752 | 1 | 0.752 | 2.086 | 0.157 |
| L_SOD | | 2307.474 | 1 | 2307.474 | 2.486 | 0.124 |
| L_CAT | P supply level (a) | 1.660 | 1 | 1.660 | 0.549 | 0.463 |
| R_MDA | | 0.004 | 1 | 0.004 | 0.033 | 0.858 |
| R_SOD | | 85230.996 | 1 | 85230.996 | 36.894[**] | 0.000 |
| R_CAT | | 12.573 | 1 | 12.573 | 8.133[**] | 0.007 |
| L_MDA | | 3.412 | 2 | 1.706 | 4.733[*] | 0.015 |
| L_SOD | | 8176.246 | 2 | 4088.123 | 4.404[*] | 0.019 |
| L_CAT | AMF inoculation (b) | 4.494 | 2 | 2.247 | 0.744 | 0.482 |
| R_MDA | | 0.147 | 2 | 0.073 | 0.600 | 0.554 |
| R_SOD | | 157418.033 | 2 | 78709.017 | 34.071[**] | 0.000 |
| R_CAT | | 8.233 | 2 | 4.116 | 2.663 | 0.083 |
| L_MDA | | 2.003 | 2 | 1.002 | 2.779 | 0.075 |
| L_SOD | | 3304.600 | 2 | 1652.300 | 1.780 | 0.183 |
| L_CAT | a × b | 0.085 | 2 | 0.042 | 0.014 | 0.986 |
| R_MDA | | 0.131 | 2 | 0.065 | 0.534 | 0.591 |
| R_SOD | | 46969.028 | 2 | 23484.514 | 10.166[**] | 0.000 |
| R_CAT | | 22.745 | 2 | 11.373 | 7.356[**] | 0.002 |
| L_MDA | | 12.976 | 36 | 0.360 | | |
| L_SOD | | 33418.629 | 36 | 928.295 | | |
| L_CAT | Error | 108.770 | 36 | 3.021 | | |
| R_MDA | | 4.410 | 36 | 0.122 | | |
| R_SOD | | 83166.315 | 36 | 2310.175 | | |
| R_CAT | | 55.653 | 36 | 1.546 | | |
| L_MDA | | 424.129 | 42 | | | |
| L_SOD | | 1514824.289 | 42 | | | |
| L_CAT | Total | 752.348 | 42 | | | |
| R_MDA | | 76.114 | 42 | | | |
| R_SOD | | 4056049.315 | 42 | | | |
| R_CAT | | 1464.697 | 42 | | | |

**Notes.**

L_MDA represents content of MDA in leaves, R_MDA represents content of MDA in roots, L_SOD represents enzyme activity of SOD in leaves, R_SOD represents enzyme activity of SOD in roots, L_CAT represents enzyme activity of CAT in leaves and R_CAT represents enzyme activity of CAT in roots.

* and ** represent the influence of factors on the indicators reached a significant difference at $p < 0.05$ and $p < 0.01$, respectively.

had higher root MDA (R_MDA) content than the CK when root inoculated with *F.m* or *R.i* in the P1 treatment. In addition, root inoculation with *R.i* reduced leaf MDA (L_MDA) content, and root inoculation with *F.m* exacerbated leaf MDA accumulation in Chinese fir during P1 treatment. The MDA content of Chinese fir leaves was significantly higher ($p < 0.05$) after root inoculation with *F.m* than inoculation with *R.i* treatment. The trend of L_MDA of Chinese fir seedlings in different treatment groups was *F.m* > CK > *R.i*, and the values of L_MDA were 3.79, 3.32, and 2.60, respectively.

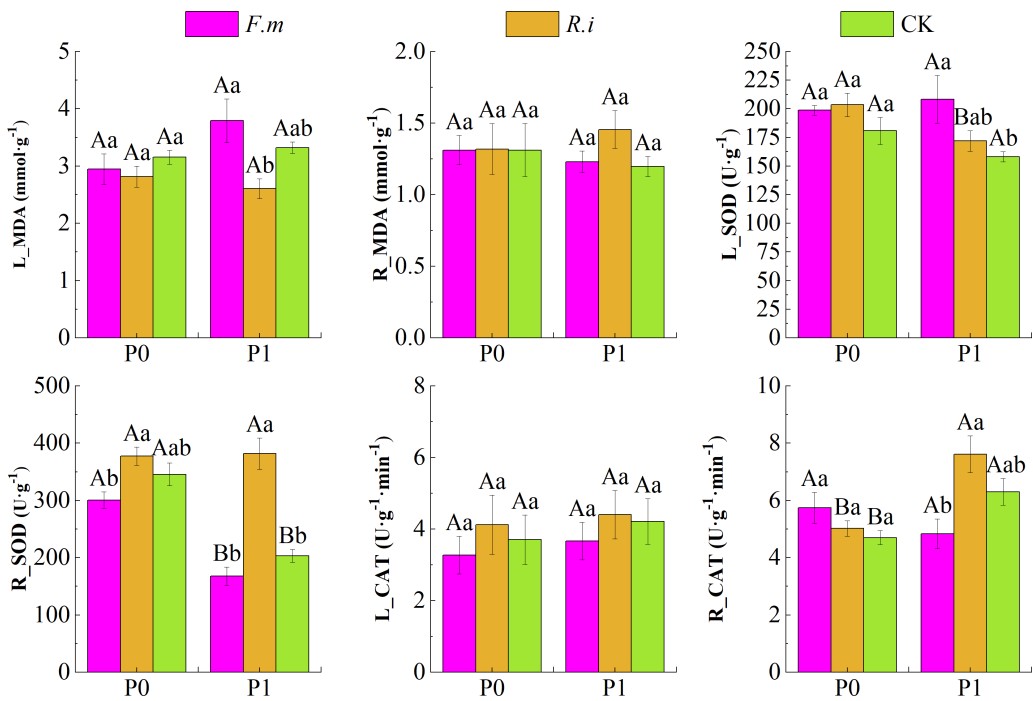

**Figure 4** **Effects of Arbuscular mycorrhizal fungi (AMF) inoculation on enzyme activities in leaves and roots of Chinese fir under different phosphorus levels.** In the figure, P0 represents no phosphorus treatment, P1 represents normal phosphorus treatment; *F.m* represents treatment with *Funneliformis mosseae* inoculation, *R.i* represents treatment with *Rhizophagus intraradices* inoculation, and CK represents the no inoculation treatment. L_MDA represents the content of MDA in leaves, R_MDA represents the content of MDA in roots, L_SOD represents the enzyme activity of SOD in leaves, R_SOD represents enzyme activity of SOD in roots, L_CAT represents enzyme activity of CAT in leaves and R_CAT represents enzyme activity of CAT in roots. Different capital letters represent a significant difference between different phosphorus treatments under the same AMF inoculation treatment. Whereas different lowercase letters represent significant differences between different AMF inoculation treatments when the phosphorus supply treatment is the same ($p < 0.05$).

Root inoculation with *F.m* or *R.i* increased leaf SOD (L_SOD) activity in both P supply treatments and root CAT (R_CAT) activity showed a similar pattern after root inoculation with *F.m* or *R.i* in the P0 treatment in Chinese fir. In particular, Chinese fir root inoculation with *F.m* at P1 treatment resulted in significantly ($p < 0.05$) higher leaf SOD activities than the CK treatment. The trend of L_SOD of Chinese fir seedlings in different treatment groups was *F.m* > *R.i* > CK, and the values of L_SOD were 208.46, 171.95, and 158.15, respectively. Under the two P supply levels, the trend of R_SOD of Chinese fir seedlings in different treatment groups was *R.i* > CK > *F.m*. Under the P1 treatment, the values of R_SOD were 381.73, 203.54, and 167.99, respectively. However, under the P0 treatment, the values of R_SOD were 377.39, 345.42, and 300.74, respectively.

In contrast, root SOD and leaf CAT (L_CAT) activities were lower than other inoculation treatments after root inoculation with *F.m*. However, root SOD and L_CAT enzyme activities were enhanced after root inoculation with *R.i*. Chinese fir showed a similar pattern of root CAT enzyme activities after root inoculation with *F.m* or *R.i* in P1 treatments.

Among them, the root SOD activity of Chinese fir was significantly higher ($p < 0.05$) than other inoculation treatments after root inoculation with *R.i* at P1 treatment, and the root CAT activity was also significantly higher ($p < 0.05$) than inoculation with *F.m* treatment. The trend of R_CAT of Chinese fir seedlings in different treatment groups was *R.i* > CK > *F.m*, and the values of R_CAT were 7.61, 6.30, and 4.84, respectively.

## Changes in endogenous hormone content in seedling leaves and roots

There was no significant interaction ($p > 0.05$) between the two factors of P supply level and AMF inoculation on endogenous hormones of Chinese fir leaves and roots. Among them, the single factors of P supply level and AMF inoculation also had no significant effect ($p > 0.05$) on endogenous hormones of Chinese fir leaves and roots (Table 4).

According to the effects of AMF inoculation on endogenous hormone content in leaves and roots of Chinese fir under different P levels (Fig. 5), root inoculation with *F.m* or *R.i* increased the contents of endogenous hormone growth hormone (IAA) in leaves and endogenous hormones abscisic acid (ABA) and zeatin riboside (ZR) in root system, and decreased the contents of endogenous hormone ABA in leaves of Chinese fir at both P supply treatments. Among them, Chinese fir root endogenous hormone IAA content was significantly higher ($p < 0.05$) than the CK treatment after root inoculation with *F.m* or *R.i* only in the P1 treatment. The trend of R_IAA of Chinese fir seedlings in different treatment groups was *F.m* > *R.i* > CK, and the values of R_IAA were 664.87, 637.50, and 571.53, respectively. Still, none of the other hormone indexes among different inoculation treatments reached a significant difference ($p > 0.05$) when Chinese fir was treated with different P supply treatments.

In addition, root inoculation with *F.m* or *R.i* increased root endogenous hormone IAA content and decreased leaf endogenous hormone ZR content in Chinese fir at P1 treatment. In the P0 treatment, the magnitude trend of endogenous hormone ZR content in Chinese fir leaves after root inoculation with *F.m* or *R.i* showed *R.i* > CK > *F.m*. In contrast, the magnitude trend of endogenous hormone IAA content in roots showed *F.m* > CK > *R.i* (Fig. 5).

## Correlation between different growth indexes of Chinese fir

According to the correlation plot between different growth indices of Chinese fir (Fig. 6), there was a significant positive correlation present between SH and fluorescence parameters ($p < 0.05$). RCD showed the opposite pattern, with a significant negative correlation between RCD and *Fv/Fm* and *Fv/Fo* ($p < 0.05$). Chl a/b had significant negative correlations with SH, L_SOD, fluorescence parameters *Fm*, *Fv/Fm*, *Fv/Fo*, and significant positive correlations with RCD ($p < 0.05$). The photochemical potential of Chinese fir leaves was enhanced when Chl a/b was low, SOD enzyme activity increased, and the growth trend of Chinese fir tended to slow down the aboveground growth to promote radial growth to resist adversity stress. Accumulation of MDA in leaves mobilized SOD enzyme activity in leaves and inhibited SOD activity in the root system. Significant positive correlations ($p < 0.05$) were found between L_SOD and L_MDA, *Fv/Fm* or *Fv/Fo*.

**Table 4 Two-way ANOVA of the effects of phosphorus supply level and arbuscular mycorrhizal fungi (AMF) inoculation on endogenous hormone content in leaves and roots of Chinese fir.**

| Index | Factor | SS | df | MS | F | p |
|---|---|---|---|---|---|---|
| L_IAA | | 9497.459 | 1 | 9497.459 | 1.210 | 0.279 |
| L_ABA | | 244.181 | 1 | 244.181 | 0.080 | 0.778 |
| L_ZR | P supply level (a) | 1.920 | 1 | 1.920 | 1.277 | 0.266 |
| R_IAA | | 795.180 | 1 | 795.180 | 0.192 | 0.664 |
| R_ABA | | 387.054 | 1 | 387.054 | 0.378 | 0.543 |
| R_ZR | | 1.786 | 1 | 1.786 | 2.015 | 0.164 |
| L_IAA | | 18139.037 | 2 | 9069.519 | 1.156 | 0.326 |
| L_ABA | | 8141.453 | 2 | 4070.726 | 1.339 | 0.275 |
| L_ZR | AMF inoculation (b) | 3.272 | 2 | 1.636 | 1.088 | 0.348 |
| R_IAA | | 16062.346 | 2 | 8031.173 | 1.937 | 0.159 |
| R_ABA | | 2502.916 | 2 | 1251.458 | 1.221 | 0.307 |
| R_ZR | | 1.755 | 2 | 0.878 | 0.990 | 0.381 |
| L_IAA | | 12615.957 | 2 | 6307.979 | 0.804 | 0.455 |
| L_ABA | | 453.282 | 2 | 226.641 | 0.075 | 0.928 |
| L_ZR | a × b | 1.947 | 2 | 0.974 | 0.648 | 0.529 |
| R_IAA | | 19110.690 | 2 | 9555.345 | 2.305 | 0.114 |
| R_ABA | | 1686.330 | 2 | 843.165 | 0.822 | 0.447 |
| R_ZR | | 1.162 | 2 | 0.581 | 0.655 | 0.525 |
| L_IAA | | 282495.412 | 36 | 7847.095 | | |
| L_ABA | | 109450.903 | 36 | 3040.303 | | |
| L_ZR | Error | 54.111 | 36 | 1.503 | | |
| R_IAA | | 149238.690 | 36 | 4145.519 | | |
| R_ABA | | 36909.274 | 36 | 1025.258 | | |
| R_ZR | | 31.904 | 36 | 0.886 | | |
| L_IAA | | 9383303.498 | 42 | | | |
| L_ABA | | 1916155.646 | 42 | | | |
| L_ZR | Total | 1121.677 | 42 | | | |
| R_IAA | | 16344704250 | 42 | | | |
| R_ABA | | 6195608.059 | 42 | | | |
| R_ZR | | 3945.822 | 42 | | | |

**Notes.**
L_IAA represents the content of IAA in leaves, R_IAA represents the content of IAA in roots, L_ABA represents the content of ABA in leaves, R_ABA represents the content of ABA in roots, L_ZR represents the content of ZR in leaves and R_ZR represents the content of ZR in roots.

## DISCUSSION

P is closely related to mycorrhizal symbionts, and different levels of P supply may promote or inhibit AMF infestation and the formation of mycorrhizal structures. It has been shown that P uptake was significantly correlated with the rate of AMF colonization with the addition of different concentrations of $KH_2PO_4$ (*Xu et al., 2014*). This study found that the colonization rate of *F.m* and *R.i* to the roots of Chinese fir at the P0 treatment was higher than that of the P1 treatment. In the P0 treatment, the colonization rate in seedling roots was the highest after inoculation with *F.m* strain, 1.66 times that of the P1 treatment

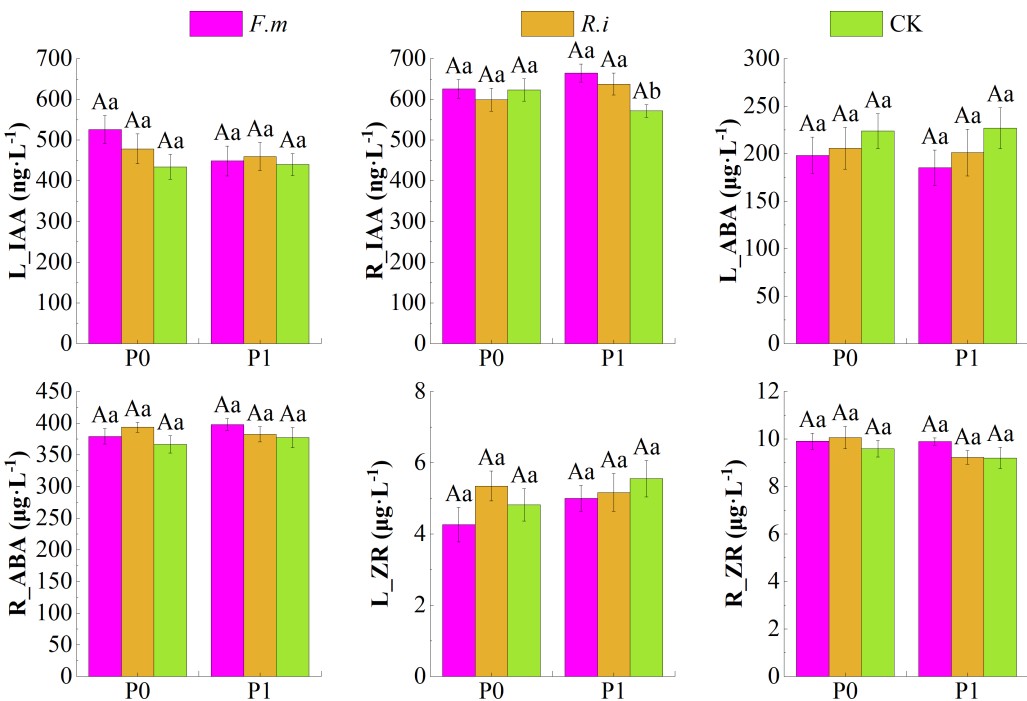

**Figure 5** **Effects of arbuscular mycorrhizal fungi (AMF) inoculation on endogenous hormone content in leaves and roots of Chinese fir under different phosphorus levels.** In the figure, P0 represents no phosphorus treatment, P1 represents normal phosphorus treatment; *F.m* represents treatment with *Funneliformis mosseae* inoculation, *R.i* represents treatment with *Rhizophagus intraradices* inoculation, and CK represents the no inoculation treatment. L_IAA represents the content of IAA in leaves, R_IAA represents the content of IAA in roots, L_ABA represents the content of ABA in leaves, R_ABA represents the content of ABA in roots, L_ZR represents the content of ZR in leaves and R_ZR represents the content of ZR in roots. Different capital letters represent a significant difference between different phosphorus treatments under the same AMF inoculation treatment, whereas different lowercase letters represent significant differences between different AMF inoculation treatments when the phosphorus supply treatment is the same ($p < 0.05$).

(Fig. 1). The second is the inoculation of *R.i* strain P0 treatment; the colonization rate in seedling roots is 1.47 times that of P1 treatment (*Tian et al., 2023*). This may be because, at normal levels of P supply, plants can take up enough P by their root extension. The need to rely on mycorrhizal fungi to take up P is less dependent. In contrast, under P deficiency, plants strengthen their symbiotic relationship with AMF and rely on the mycelial network of AMF to help them obtain enough P to sustain their growth (*Baird, Walley & Shirtliffe, 2010*; *Xie et al., 2014*).

By monitoring the growth of seedling height and RCD of Chinese fir root inoculated with AMF, it was found that AMF could promote seedling height growth at both levels of P supply, and *R.i* had a stronger effect on seedling height than *F.m* (Fig. 2). Under abundant P supply, AMF inhibited the growth of RCD of Chinese fir. Still, nutrient scarcity inhibited the growth of RCD of Chinese fir in the case of P deficiency. Then AMF could reduce this inhibitory effect, and the ability of *R.i* to reduce this effect was stronger than *F.m*. The promotion of AMF on plant morphological growth is closely related to the response of
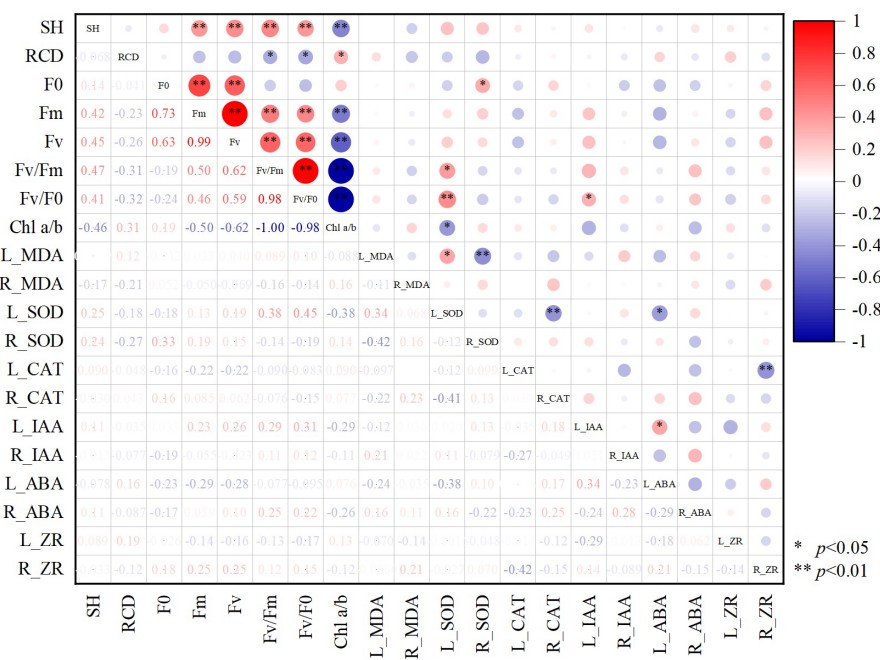

**Figure 6** **The correlation between different growth indexes of Chinese fir.** SH represents seedling height increment and RCD represents root collar diameter increment. *Fo* represents origin fluorescence, *Fv* represents variable fluorescence, *Fm* represents maximum fluorescence, *Fv/Fm* represents maximal photochemical efficiency of PSII, *Fv/Fo* represents the potential photochemical activity of PSII, Chl a/b represents the ratio of chlorophyll-a content to chlorophyll b content. L_MDA represents the content of MDA in leaves, R_MDA represents the content of MDA in roots, L_SOD represents the enzyme activity of SOD in leaves, R_SOD represents enzyme activity of SOD in roots, L_CAT represents enzyme activity of CAT in leaves, R_CAT represents enzyme activity of CAT in roots. L_IAA represents the content of IAA in leaves, R_IAA represents the content of IAA in roots, L_ABA represents the content of ABA in leaves, R_ABA represents the content of ABA in roots, L_ZR represents the content of ZR in leaves, R_ZR represents the content of ZR in roots. The red color represents a positive correlation between the two indicators, and blue represents a negative correlation between the two indicators, and the depth of the color represents the level of correlation. An asterisk (*) represents that the correlation between the two indicators reaches a significant difference level ($p < 0.05$), two asterisks (**) represent that the correlation between the two indicators reaches a very significant difference level ($p < 0.01$).

AMF to host nutrient acquisition capacity, photosynthesis enhancement and physiological and metabolic levels-mycorrhizal symbiosis as a key component to help plants survive under unfavorable environmental conditions (*Sawers, Gutjahr & Paszkowski, 2008*). The symbiotic relationship between AMF and the host is mainly carried out through the mycelium. When AMF helps the plant absorb the mineral nutrients in the soil through the mycelium, the plant will also return a part of its photosynthesis-generated products to AMF through the mycelium to ensure a sustainable and mutually beneficial symbiotic partnership between them (*Parniske, 2008*).

AMF can enhance plant photosynthetic productivity by regulating plant chlorophyll content and improving chlorophyll fluorescence properties, such as inhibiting *Fo* and increasing *Fv/Fm* and *Fv/Fo* during plant photosynthesis so that the symbiotic plants can better utilize the limited resources in unfavorable environments to sustain growth and AMF

nutrients (*Zai et al., 2012*). In the course of Chl fluorescence analysis, *Fv/Fm* and *Fv/Fo*, as measures of the primary photochemical capacity of PSII, can reflect the photochemical activity and optical properties of leaves, and PSII itself is particularly sensitive to various environmental stress inducers (*Baker & Rosenqvist, 2004*; *Henriques, 2009*). The results of the present study revealed that P deficiency attenuated *Fv/Fm* and *Fv/Fo* of Chinese fir to varying degrees. Lower *Fv/Fm* values indicate that a part of the PSII reaction center is damaged or inactivated, a common phenomenon in plants under stress. P deficiency may limit electron transfer from the PSII receptor side (*Shu et al., 2012*).

Inoculation of Chinese fir roots with AMF improved the chlorophyll fluorescence characteristics of leaf *Fv*, *Fm*, *Fv/Fm*, and *Fv/Fo* (Fig. 3). AMF reduced the leaf fluorescence characteristic *Fo* and enhanced the leaf PSII photochemical efficiency to different extents at different levels of P supply treatment (*Zai et al., 2012*). The enhancement of leaf PSII photochemical efficiency by AMF was stronger in P deficiency. AMF also reduced Chl a/b in leaves, which may be after AMF reached symbiotic dependence with Chinese fir, through some metabolites regulating the phototropism of the plant, and reduced Chl a/b in leaves; in general, AM plants showed higher chlorophyll levels than non-AM plants, but under nutrient stress plants less nutrients will inevitably affect the level of chlorophyll production, limited nutrient resources may have been spent on the most appropriate regulatory processes for Chl a/b in the current habitat, ensuring that plants can carry out the production of photosynthetically produced products relatively efficiently even under low light levels (*Marschall & Proctor, 2004*; *Sannazzaro et al., 2006*; *Hernández & Munné-Bosch, 2015*). This may be more closely related to the mechanism by which AMF can enhance the photochemical efficiency of leaf PSII.

AMF can cope with environmental stresses by establishing a symbiotic relationship with the host and exchanging nutrients with the host while mobilizing enzyme metabolism systems in the plant to improve plant resistance (*Wang et al., 2023*). Although no study has shown that AMF can reduce the persecutory effects of P deficiency on plant growth, the AMF-inoculated plants in this study will have lower levels of MDA and higher protease activities relative to plants without AMF symbiosis, and *R.i*-inoculated Chinese fir will have higher SOD and CAT enzyme activities relative to *F.m*.

The leading cause of stress-induced decrease in plant productivity is redox imbalance and oxidative damage at the cellular level. The enzymatic defense system in plants can actively increase the activity of enzymes, such as SOD and CAT, to enhance the elimination of toxic metabolites from the cells in the presence of redox imbalance and oxidative damage (*Allen, 1995*; *Kandlbinder et al., 2004*). In the experimental results, the activities of SOD and CAT enzymes were enhanced to different degrees in AMF-inoculated plants relative to those without AMF symbiosis. This confirms that AMF can improve plant stress tolerance during nutrient deprivation or environmental stress (*Begum, Ahanger & Zhang, 2020*). However, exactly which substances are involved in this intrinsic regulatory process in plants is unknown.

Hormone balance plays different regulatory roles when plants are subjected to environmental stresses, and growth regulation of plants is mainly exerted through the reciprocal effects of different hormones (*Khan et al., 2013*). Some scholars have shown that

AMF increased the secretion of the endogenous plant hormone IAA, significantly increased the expression levels of *NADP-ME1* and *NADP-ME2* and the activity of *NADP-ME*, and enhanced the root activity of tomato (*Wang et al., 2021*). We measured the content of endogenous hormones in Chinese fir and found that there were differences in the levels of hormone content in aboveground parts, such as Chinese fir leaves and root parts, in which the endogenous hormone IAA content in leaves was higher in Chinese fir with AMF inoculation than in Chinese fir without AMF symbiosis regardless of any level of P supply (Fig. 5).

Still, there was no significant response for the endogenous hormone IAA content in the root system. Only when the P supply was sufficient, AMF increased IAA content in roots. IAA promotes the growth of aboveground height and leaf extension and also promotes the distribution of roots to regulate the growth process of plants (*Zhao, 2010*). The symbiotic mechanism between AMF and its host may be adjusted by environmental P concentration, which follows the optimal foraging theory. When nutrients are abundant, the plant enhances inter-root nutrient uptake and utilization by allocating more signals and substances to the underground root system to promote root proliferation.

In contrast, when nutrients are scarce, AMF helps the plant to forage for nutrients from the farther soil environment at a lower cost of carbon inputs through mycelial network expansion (*Johnson, 2010*). The endogenous hormone ABA plays a major role in regulating root growth, and ABA controls both root ABA levels and root growth synthesized in leaves rather than from roots. ABA synthesized in leaves can be transported to roots, which may act as a signal to promote root growth (*McAdam, Brodribb & Ross, 2016*).

ZR, as a class of cytokinins, is complementary to the five major plant hormones, and cytokinins also play an important role in root development and structure (*Hodge et al., 2009*). The present study found that endogenous hormones ABA and ZR content in the plant root system would be higher in Chinese fir with AMF inoculation than in Chinese fir without AMF symbiosis (Fig. 5). This response was reflected in the opposite way within the leaves. The inhibitory effect of ABA on branch growth was evident and pervasive and is thought to arise mainly from the induction of stomatal closure and assimilation of reduced (*Tardieu, Parent & Simonneau, 2010*). AMF promotes the flow of endogenous hormones from leaves to the plant root system, which promotes root growth. Limited resources are used primarily to balance plant aboveground and below-ground growth relationships in nutrient scarcity. In addition, the colonization of plant roots by AMF mainly occurred in such young roots as lateral roots, and the epidermis of young roots may be more susceptible to mycelial penetration and infestation (*Chabaud et al., 2002*). The flow of hormones contributing to root growth from leaves to roots also proved that AMF might induce plant roots to branch out more lateral roots to enhance mycorrhizal symbiosis through this pathway.

On the other hand, the results showed that the differences in the effects of *F.m* and *R.i* strains on endogenous hormone content were related to P content, with *R.i* having a higher regulatory capacity to enhance endogenous hormone content in Chinese fir leaves relative to *F.m* when P was in a sufficient supply. The opposite was true in the root system. In contrast, when P was deficient, the content of endogenous hormone IAA was

higher in both *F.m*-inoculated Chinese fir leaves and root system than in *R.i*-inoculated Chinese fir. However, *R.i*-inoculated Chinese fir possessed a higher content of endogenous hormones ABA and ZR relative to *F.m* leaves and root system. This may be related to the variability of the main symbiotic strategies of different AMF species towards their hosts at different nutrient concentrations (*Qin et al., 2017*). When P nutrient deficiencies limit plant growth, the benefits of mutual support between AMF and the host are realized (*Hoeksema et al., 2010*). The present study also found that although *F.m* and *R.i* strains had high and low levels of host regulation at different nutrient concentrations, this difference was insignificant ($p < 0.05$).

This may be because the differences in growth effects of different AMF taxa are mainly reflected at the genus level rather than the species level (*Hart & Klironomos, 2003*). However, the AMF taxa used in this study belonged to the same genus (*Glomus*), and it was found that complementary effects of AMF diversity were more likely to be found when different AMF genera (with different strategies) were present. Therefore, future studies testing whether AMF diversity promotes plant productivity should include AMF taxa from different genera (*Van Der Heijden et al., 2006*). In addition, natural forests composed of multiple species have higher soil quality and fungal abundance than plantation forests composed of a single species (*Guo et al., 2022*).

Moreover, most AMFs are not host-specific, and their mycelium can simultaneously infect the root systems of different plants. Various AMFs can also infect the same plant, and the root mycelium can fuse, eventually forming complex arbuscular mycorrhizal networks (*Simard et al., 2012*). Therefore, when researching the application of AMF in forestry, the experimental object should be shifted from the study of a single AMF species on a single tree species to the study of the mycorrhizal networks formed between multiple AMF species and different tree species to truly understand the role of AMF and forest trees in the natural environment, and to provide theoretical guidance for the entire play of productivity of plantation forests.

## CONCLUSIONS

Under different P supply treatments, root inoculation of AMF promoted SH growth, and the promotion effect of *R.i* on SH growth was greater than that of *F.m*. In P1 treatment, root inoculation of AMF showed an inhibitory effect on the RCD growth of Chinese fir, and the inhibitory effect of *R.i* on the RCD growth was greater than that of *F.m*. When P was deficient, the growth of Chinese fir RCD was weakened. At this time, inoculation with *F.m* or *R.i* promoted the growth of Chinese fir RCD. The promotion effect of *R.i* on RCD growth was more substantial than that of *F.m*. Inoculation of *R.i* or *F.m* could help Chinese fir significantly enhance its adaptive capacity to the low P environment by regulating the fluorescence response of Chl, mainly promoting the maximum photochemical activity of PSII, stimulating the activity of antioxidant enzymes, and promoting the high and radial growth of fir by regulating the hormone balance above ground and in the root system, which significantly enhanced the adaptive ability of Chinese fir to the low P environment. The colonization of AMF in plant roots can benefit forestry management and development

by improving P utilization efficiency, promoting plant growth and development, and improving stress resistance and ecological adaptability. This reduces operating costs, improves production efficiency, and contributes to the sustainable use and protection of the environment. In the future, combining the plant phenotypic growth indexes with molecular biology is suggested to analyze the expression levels of the relevant genes in the physiological processes to deeply analyze the differences in the interactions between different AMFs and their hosts.

**Abbreviations**

| | |
|---|---|
| **AMF** | Arbuscular mycorrhizal fungi |
| **SH** | Seedling height increment |
| **RCD** | Root collar diameter increment |
| **L_MDA** | Content of MDA in leaves |
| **R_MDA** | Content of MDA in roots |
| **L_SOD** | Enzyme activity of SOD in leaves |
| **R_SOD** | Enzyme activity of SOD in roots |
| **L_CAT** | Enzyme activity of CAT in leaves |
| **R_CAT** | Enzyme activity of CAT in roots |
| **L_IAA** | Content of IAA in leaves |
| **R_IAA** | Content of IAA in roots |
| **L_ABA** | Content of ABA in leaves |
| **R_ABA** | Content of ABA in roots |
| **L_ZR** | Content of ZR in leaves |
| **R_ZR** | Content of ZR in roots |

# ACKNOWLEDGEMENTS

We are very grateful to the editors, reviewers and experts who made suggestions for the manuscript.

## Funding

The study was funded by the National Key Research and Development Program of China (2021YFD2201304-05), the Forestry Science and Technology Program of Fujian Province, China (2023FKJ09), and the Professional Degree Postgraduate Course Teaching Case Library Construction Project of Fujian Agriculture and Forestry University (712018270478). The funders had no role in study design, data collection and analysis, decision to publish, or preparation of the manuscript.

## Grant Disclosures

The following grant information was disclosed by the authors:
National Key Research and Development Program of China: 2021YFD2201304-05.
Forestry Science and Technology Program of Fujian Province, China: 2023FKJ09.
Professional Degree Postgraduate Course Teaching Case Library Construction Project of Fujian Agriculture and Forestry University: 712018270478.

## Competing Interests

The authors declare there are no competing interests.

## Author Contributions

- Yunlong Tian conceived and designed the experiments, performed the experiments, analyzed the data, prepared figures and/or tables, and approved the final draft.
- Jingjing Xu performed the experiments, prepared figures and/or tables, and approved the final draft.
- Linxin Li performed the experiments, prepared figures and/or tables, and approved the final draft.
- Taimoor Hassan Farooq analyzed the data, prepared figures and/or tables, authored or reviewed drafts of the article, and approved the final draft.
- Xiangqing Ma conceived and designed the experiments, authored or reviewed drafts of the article, and approved the final draft.
- Pengfei Wu conceived and designed the experiments, authored or reviewed drafts of the article, and approved the final draft.

## Data Availability

The raw data is available in the Supplementary File.

## Supplemental Information

Supplemental information for this article can be found online at http://dx.doi.org/10.7717/peerj.17138#supplemental-information.

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
