# Peer review of "Effect of arbuscular mycorrhizal symbiosis on growth and biochemical characteristics of Chinese fir (Cunninghamia lanceolata) seedlings under low phosphorus environment"

_PeerJ, doi:10.7717/peerj.17138_

## Round 0.1 · original submission · Major Revisions

The authors need to include all suggestions given by both reviewers.

**Language Note:** PeerJ staff have identified that the English language needs to be improved. When you prepare your next revision, please either (i) have a colleague who is proficient in English and familiar with the subject matter review your manuscript, or (ii) contact a professional editing service to review your manuscript. PeerJ can provide language editing services - you can contact us at copyediting@peerj.com for pricing (be sure to provide your manuscript number and title). – PeerJ Staff

Reviewer 1 ·

Basic reporting

no comment

Experimental design

no comment

Validity of the findings

no comment

Additional comments

Specific Comments:
Abstract. Please clarify the rationale behind choosing one-and-half-year-old seedlings and the significance of this age group to the study's objectives. Also, please specify the quantitative results obtained for each treatment group (P0, P1, CK) to allow readers to comprehend the magnitude of the observed effects better. Include statistical measures or significance levels where applicable.
Line 54. Quantify statements when possible. For instance, instead of stating "up to 20%", provide a specific range or reference for the carbon transfer from plants to AMF.
Line 111. What does Chinese fir artificial productivity mean? Please clarify.
Lines 118-125. In the sentence "Then, how does the growth stress response process of Chinese fir behave after AMF forms a symbiotic relationship with Chinese fir?", consider rephrasing for clarity and grammatical correctness.
Lines 118-125. The hypotheses are mentioned but could be more clearly outlined. Consider separating them into distinct points and elaborating on the expected outcomes.
Lines 126-133. This description should belong to the Materials and Methods section.
Conclusions. There is a lack of discussion on the potential practical implications of the study, such as potential benefits for agriculture or forestry.

Reviewer 2 ·

Basic reporting

no comment

Experimental design

no comment

Validity of the findings

no comment

Additional comments

This paper aimed to investigate the effects of Arbuscular mycorrhizal fungi (AMF) on the
growth and physiological characteristics of Chinese fir under different P supply treatments. The
structure of the paper is complete, and underscore the positive impact of AMF, particularly Gi
inoculation, on various aspects of Chinese fir seedling growth and stress response, shedding light
on its potential role in optimizing plant health under different conditions. Therefore, min or
revisions (such as descriptions in detail on materials and methods, check the data and the
statement of results and conclusion , describe the results in further detail, check the format of the
references ) are needed before acceptance . Some suggested amendments are as follows:

(1) Line 55, “...up to 20 % of the plant's fixed carbon is transferred to AMF via mycelium ”, the
allocation of photosynthetic carbon to AMF by plants ranges from 4% to 25%, rather than
being a fixed value. Suggested authors to revised in this context.

(2) Line 145-146, the duration of high-temperature sterilization is not specified. Whether it is
twenty minutes or two hours is not mentioned.

(3) Line 149-150, the nomenclature of AMF has been updated since 2019, and Glomus mosseae
and Glomus intraradices have been reclassified as Funneliformis mosseae and Rhizophagus
intraradices, respectively. I recommend that the authors update the fungal names accordingly
and provide appropriate references supporting this modification.

(4) Line 170, this section's logic is incorrect; photosynthetic parameters should be measured first,
followed by the assessment of growth parameters, and finally, samples should be collected
for the determination of other indicators.

(5) Line 175, what is “tufts ”? It should be arbuscular.

(6) Line 178, “infestation rates ” should be colonization.

(7) Line 180, it is recommended that the authors supplement relevant images depicting the
colonization. At the same time, this portion of the content should be presented in the form of
results.

(8) Line 186-189, the methods for detecting MDA, CAT, and NBT are recommended to be
further refined.

(9) In the Results section, it is suggested that the authors present the variations in data, not just
the significant changes.

(10) It is suggested that the authors supplement the results of the two- way ANOVA in the figures
and tables.

(11) The number of references from the last five years is relatively low. It is suggested that the
authors supplement their references, particularly with publications from 2020 onwards.

---

## Round 0.2 · Minor Revisions

The authors need to keep respective figure number in Discussion section while providing results. Authors need to check all references carefully in the text.

Reviewer 2 ·

Basic reporting

no comment

Experimental design

no comment

Validity of the findings

no comment

Additional comments

no comment

Annotated reviews are not available for download in order to protect the identity of reviewers who chose to remain anonymous.

---

## Round 0.3 · Minor Revisions

The authors need to include all suggestions given in attached manuscript PDF.

---

## Round 0.4 · Minor Revisions

Provide the full name of abbreviations (in footnote of tables or figure) used in figure or tables (in Figure 6, Table 1-4). In figure 6, write clearly blue and red color in heat map correlation figure indicate for what information (Figure 6).

Reviewer 2 ·

Basic reporting

no comment

Experimental design

no comment

Validity of the findings

no comment

Additional comments

no comment

---

## Round 0.5 · accepted · Accept

Check all figures, tables and references carefully.